# Tuneable near white-emissive two-dimensional covalent organic frameworks

Xing Li[1,2,3], Qiang Gao[1], Juefan Wang [2,4], Yifeng Chen[2,4], Zhi-Hui Chen[1], Hai-Sen Xu[1], Wei Tang[5], Kai Leng[1,2], Guo-Hong Ning [1], Jishan Wu [1], Qing-Hua Xu [1], Su Ying Quek[2,4], Yixin Lu[1] & Kian Ping Loh[1,2]

Most two-dimensional (2D) covalent organic frameworks (COFs) are non-fluorescent in the solid state even when they are constructed from emissive building blocks. The fluorescence quenching is usually attributed to non-irradiative rotation-related or π–π stacking-caused thermal energy dissipation process. Currently there is a lack of guiding principle on how to design fluorescent, solid-state material made of COF. Herein, we demonstrate that the eclipsed stacking structure of 2D COFs can be used to turn on, and tune, the solid-state photoluminescence from non-emissive building blocks by the restriction of intramolecular bond rotation via intralayer and interlayer hydrogen bonds among highly organized layers in the eclipse-stacked COFs. Our COFs serve as a platform whereby the size of the conjugated linkers and side-chain functionalities can be varied, rendering the emission colour-tuneable from blue to yellow and even white. This work provides a guide to design new solid-state emitters using COFs.

---

[1] Department of Chemistry, National University of Singapore, 3 Science Drive 3, Singapore 117543, Singapore. [2] Centre for Advanced 2D Materials, National University of Singapore, 6 Science Drive 2, Singapore 117542, Singapore. [3] NUS Graduate School for Integrative Sciences and Engineering, National University of Singapore, Centre for Life Sciences, #05-01, 28 Medical Drive, Singapore 117456, Singapore. [4] Department of Physics, National University of Singapore, 2 Science Drive 3, Singapore 117551, Singapore. [5] Institute of Materials Research and Engineering, A*STAR, 2 Fusionopolis Way, Innovis, Singapore 138634, Singapore. Correspondence and requests for materials should be addressed to K.P.L. (email: chmlohkp@nus.edu.sg)

Covalent organic frameworks (COFs) allow molecular building blocks to be integrated into extended architectures via covalent bonds[1–5]. Guided by reticular chemistry, tremendous efforts have been directed towards developing highly crystalline COFs with different topologies such as two-dimensional (2D)[6–13], three-dimensional (3D)[14,15] and woven structures[16,17]. Meanwhile, the ease of encoding functionalities into building units makes COFs useful for wide-ranging applications such as sensing[18–22], gas storage[23,24], photocatalysis[25], asymmetric catalysis[26–29], chemical separation[30,31], energy storage[32–35] and photodetection[36]. However, most fields view COFs as merely a polymer platform and overlook the unique features of COFs such as their periodic layered structures, and this has given rise to the non-differentiated performance of COFs, amorphous polymers and even their building units.

In layered materials, π–π stacking leads to aggregation, which causes photoluminescence (PL) quenching. Previously, highly emissive building units, such as pyrene and tetraphenylethene, have been used to construct solid-state photoluminescent 2D COFs[19,37–39]. When an aggregation-induced emission (AIE) chromophore such as tetraphenylethene was connected via a boronate ester, a highly emissive 2D COF was obtained. Unfortunately, the solid-state PL was quenched when the AIE building unit was linked by imine bonds due to its rotationally labile nature under photoexcitation[40]. A similar phenomenon occurred when 2D COFs were constructed with non-AIE chromophores such as pyrene. In fact, the type of linker between the chromophores in the 2D COF strongly affects its ability to be fluorescent. Thermal dissipation via rotationally labile linkages, for example, the commonly used imine linkages, induces non-radiative decay of excited states[40]. Therefore, it is vital to restrict the bond rotation of the 2D COF linkages to induce solid-state PL.

On the other hand, 2D COFs provide the equivalent of a high-viscosity microenvironment that limits rotational/vibrational relaxation. When properly designed, the eclipsed stacking structure of 2D COFs can block the non-radiative energy decay pathway through restriction of intramolecular bond rotation (RIR), thereby enhancing fluorescence. In addition, the molecular building blocks can be interconnected as donor–acceptor pairs (dyads) that are not conjugated, or the conjugation in the scaffold can be extended using aromatic linkers, leading to donor–acceptor excimer energy transfer and a large Stokes shift, respectively. The combination of these factors can be used to tune the intensity and energy of the fluorescence, but their effects have not been studied systematically in COFs.

To date, there is a lack of guiding principles on how solid-state photoluminescent COFs with tuneable emissive properties can be constructed. Here, we demonstrate a molecular design strategy for obtaining solid-state PL from a hydrazone-based COF that allows strong emission intensity and wide colour tuneability. Starting from a non-emissive building unit (Fig. 1a), PL can be induced via the RIR mechanism when intramolecular hydrogen bonding is introduced (Fig. 1b). Taking advantage of the unique eclipsed stacking of 2D COFs, the RIR mechanism can be further reinforced by intra- and interlayer hydrogen bonding (Fig. 1c). Most interestingly, we found that the interplay between intralayer and interlayer hydrogen bonding can be used to tune the wavelength of emission, which provides a new strategy for achieving white light emission from COFs. In analogous eclipsed-stacked structure without hydrogen bonding, the fluorescence is much weaker (Fig. 1e).

Our design strategy has the following salient features. First, we start from a non-emissive hydrazone-based building unit; the fluorescence of which is quenched in the amorphous solid state due to free bond rotation. Second, intra- and interlayer hydrogen bonding and eclipsed stacking in the COF work together to reinforce RIR and enhance fluorescence. The hydrazone unit has a proton donor (N–NH) and proton acceptor (C–O–C) (both intra- and interlayer) in close proximity, which means that the

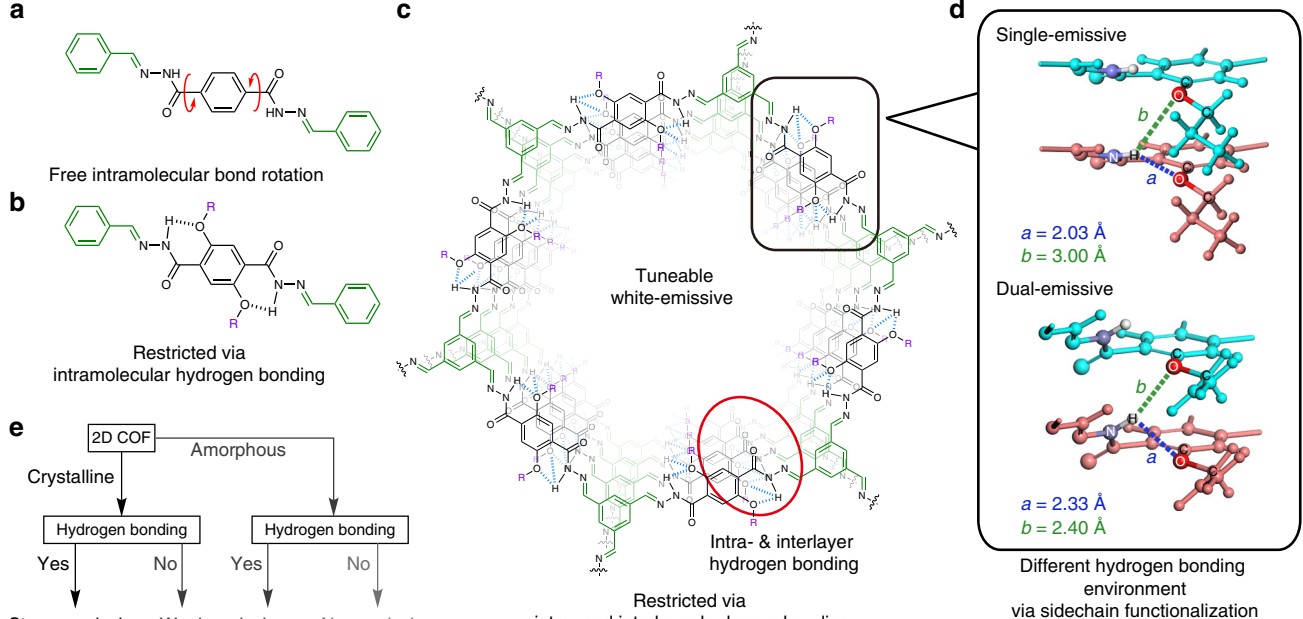

**Fig. 1** Design principles of the tuneable white-emissive 2D COFs. **a** Intramolecular rotation in the amorphous solid state makes it non-emissive. **b** Intramolecular rotation is restricted via intramolecular hydrogen bonding, leading to enhanced emission in the UV to violet region. **c** The small building units are extended into 2D COFs and immobilized via intra- and interlayer hydrogen bonding; thus, the units display further enhanced PL in the visible region. **d** Intralayer hydrogen bonding competes with interlayer bonding; single emission is observed in COFs with intra- and interlayer hydrogen bonding of 2.03 Å and 3.00 Å, while dual emission is observed in COFs with intra- and interlayer hydrogen bonding of 2.33 Å and 2.40 Å, respectively (upper layer: light blue; lower layer: pink; O: red; N: violet; H: white; intralayer hydrogen bonding: blue dashed line; and interlayer hydrogen bonding: green dashed line). **e** Flow chart to achieve strong emission in the solid state of 2D COFs

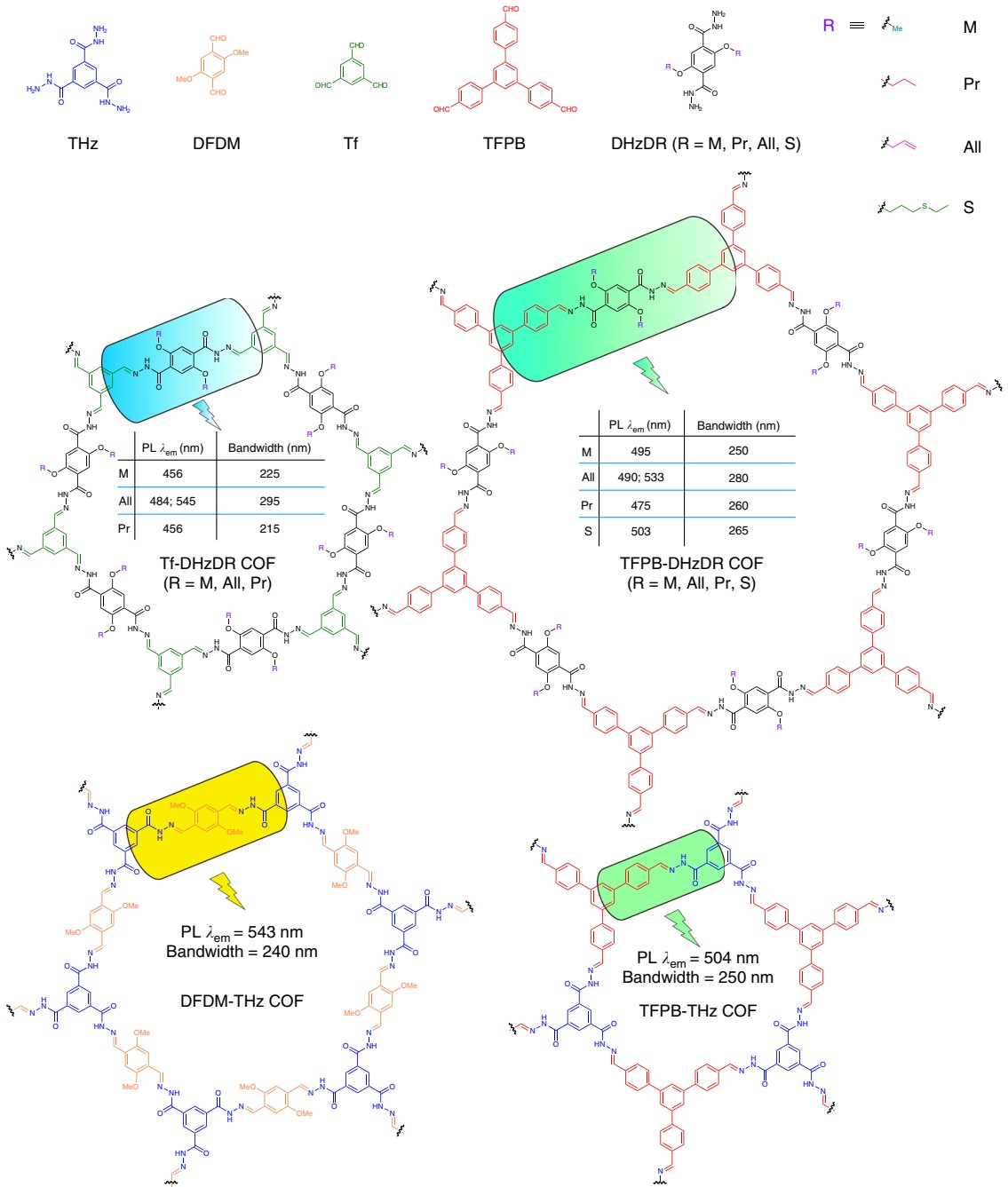

**Fig. 2** Synthetic scheme for accessing the tuneable white-emissive COFs. The photoluminescent emission maxima and bandwidths are shown for each COF

intra- and interlayer hydrogen bonds can form the basis of an excited-state proton shift mechanism. Third, a Stokes shift can be induced by incorporating a conjugated linker. Fourth, dual emission with a large Stokes shift is obtained by modulating the intra- and interlayer hydrogen bonding by adding side-chain functionalities (Fig. 1d). Last but not least, the emission colour can be further fine-tuned by construction of multicomponent COFs. Such a platform affords widely tuneable PL properties and facilitates structure–property correlation studies. For example, we discovered a distinct dual emission when an allyl group is added to the side chain, which is the first example of near white emission from COFs. Although intralayer hydrogen bonding has previously been explored for synthesis of highly crystalline COFs for photocatalysis[25], the interplay between intralayer and interlayer hydrogen bonding in eclipsed stacked COFs has never been

studied. Based on density functional theory (DFT) simulation studies, the varying degrees of intralayer and interlayer hydrogen bonding affects whether the COFs is single or dual-emissive (Fig. 1d and Supplementary Figs. 31 and 32). Accordingly, we proposed a unique COF-triggered excited-state interlayer proton shift (ESIPS) mechanism that is analogous to the excited-state intramolecular proton transfer (ESIPT)[41,42] process. Instead of the enol-keto tautomers in ESIPT, there exist two conformers with different interlayer–intralayer hydrogen environments which can give rise to the dual emission.

## Results

**Design of tuneable white-emissive COFs**. The synthetic scheme for accessing the tuneable white-emissive COFs is illustrated in

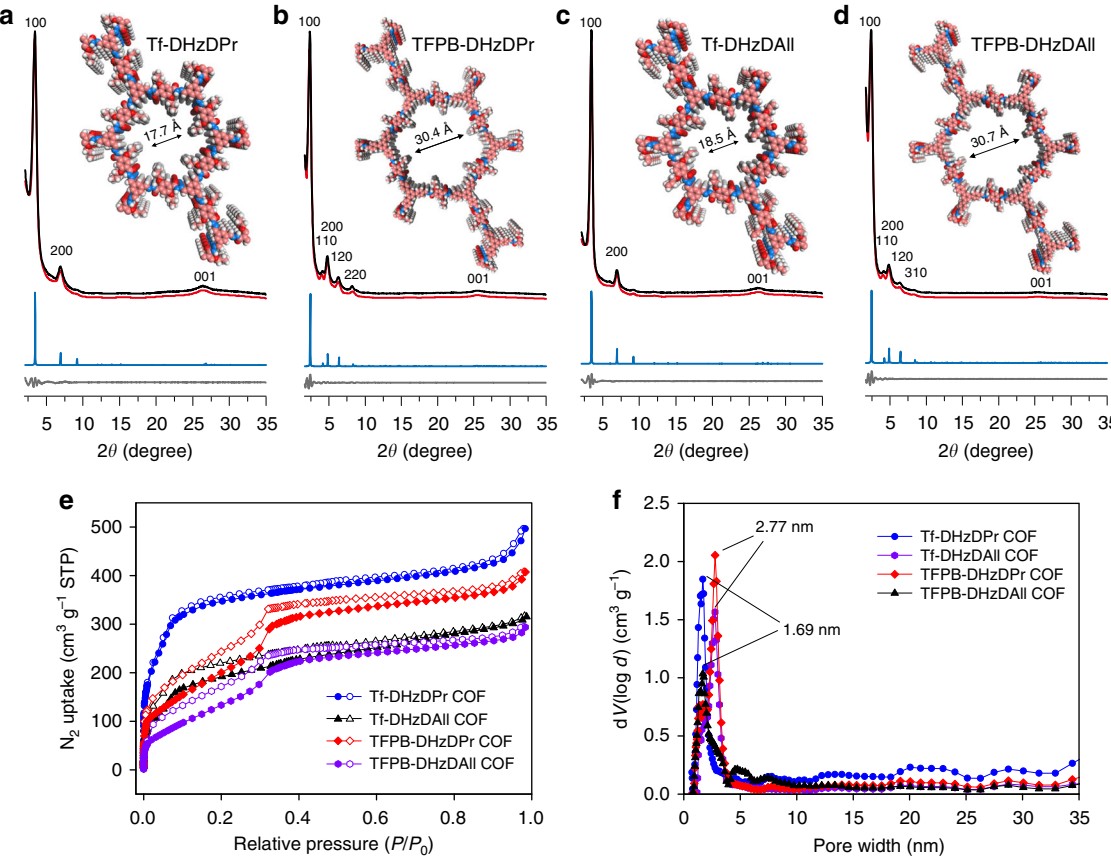

**Fig. 3** Characterization of selected COFs. Experimental (black), Pawley refined (red), their difference (grey) and simulated eclipsed stacking (blue) PXRD patterns of **a** Tf-DHzDPr COF, **b** Tf-DHzDAll COF, **c** TFPB-DHzDPr COF and **d** TFPB-DHzDAll COF. (Inset: refined eclipse structures with simulated pore sizes. C, pink; N; blue; O, red; and H, white.) **e** Nitrogen sorption isotherms. **f** Pore size distribution of the COFs

Fig. 2 and Supplementary Fig. 1. A series of nine COFs with different pore sizes and side-chain functionalities were designed and synthesized. We used three different aldehydes, 2,5-dimethoxyterephthalaldehyde (DFDM), benzene-1,3,5-tricarboxaldehyde (Tf) and 1,3,5-tri(4-formylphenyl)benzene (TFPB), as spacers to control the size of the conjugated intra-plane π-system in the 2D COFs. We also designed two different types of hydrazides, benzene-1,3,5-tricarbohydrazide (THz) and 2,5-disubstituted terephthalohydrazide (DHzDR), to elucidate the role of hydrogen bonding in restricting bond rotation. The side-chain functionality of the DHzDR was varied from methoxy, propoxy, allyloxy to 3-(ethylthio)propoxy to tune the ESIPS and PL efficiency.

**Characterization of the COFs**. The as-synthesized COFs were characterized by powder X-ray diffraction (PXRD), nitrogen sorption isotherms, FT-IR spectroscopy and elemental analysis (see Supplementary Methods and Supplementary Figs. 2–10, 13–15). Four representative COF structures, Tf-DHzDPr, Tf-DHzDAll, TFPB-DHzDPr and TFPB-DHzDAll, with similar side-chain lengths (three carbons) and different spacer sizes are discussed here. As shown in Fig. 3a, the PXRD patterns exhibit the most intense (100) peaks at 3.46° (FWHM = 0.79°), 3.46° (FWHM = 0.41°), 2.36° (FWHM = 0.67°) and 2.39° (FWHM = 0.57°) for Tf-DHzDPr, Tf-DHzDAll, TFPB-DHzDPr and TFPB-DHzDAll, respectively, indicating that all four COFs are highly crystalline. Tf-DHzDPr exhibits two additional peaks at 6.92 and 26.39°, corresponding to the (200) and (001) facets; Tf-DHzDAll displays two peaks at 6.95 and 26.26°, corresponding to the (200)

and (001) facets; TFPB-DHzDPr shows five additional peaks at 4.07, 4.80, 6.29, 8.16 and 25.60°, corresponding to the (110), (200), (120), (220) and (001) facets, respectively; TFPB-DHzDPr reveals five additional peaks at 4.13, 4.83, 6.42, 8.26 and 25.38°, corresponding to the (110), (200), (120), (310) and (001) facets. The Pawley refinement (Material Studio 2016) displays good consistency with the experimental PXRD patterns with ($R_p$, $R_{wp}$) values of (2.25%, 2.94%), (3.51%, 4.55%), (4.23%, 5.18%) and (2.69%, 3.21%) for Tf-DHzDPr, Tf-DHzDAll, TFPB-DHzDPr and TFPB-DHzDAll, respectively. As shown in Fig. 2a–d, all four COFs adopted eclipsed stacking modes belonging to the $P3$ space group with Pawley refined cell parameters of $a = b = 29.37$ Å, $c = 3.44$ Å for Tf-DHzDPr; $a = b = 29.35$ Å, $c = 3.43$ Å for Tf-DHzDAll; $a = b = 42.28$ Å, $c = 3.56$ Å for TFPB-DHzDPr; and $a = b = 41.85$ Å, $c = 3.50$ Å for TFPB-DHzDAll. Simulation of the staggered stacking mode of the same space group was also conducted for each COF, but the calculated PXRD pattern was not consistent with the experimental result (Supplementary Figs. 2–10). The fractional atomistic coordinates for Pawley-refined unit parameters of each COFs are shown in Supplementary Tables 8–16.

The porous structures of the COFs were further investigated by nitrogen adsorption–desorption experiments at 77 K (Fig. 3e). The sorption isotherms of Tf-DHzDPr and Tf-DHzDAll exhibit type-I adsorption profiles with abrupt increases at $P/P_0 < 0.1$, indicating that they are microporous materials. TFPB-DHzDPr and TFPB-DHzDAll display type-IV adsorption profiles with dramatic increases at $P/P_0 < 0.1$ and between 0.3 and 0.35, which are typical of mesoporous materials. Furthermore, the Brunauer−Emmett−Teller (BET) surface areas are 1035, 760,

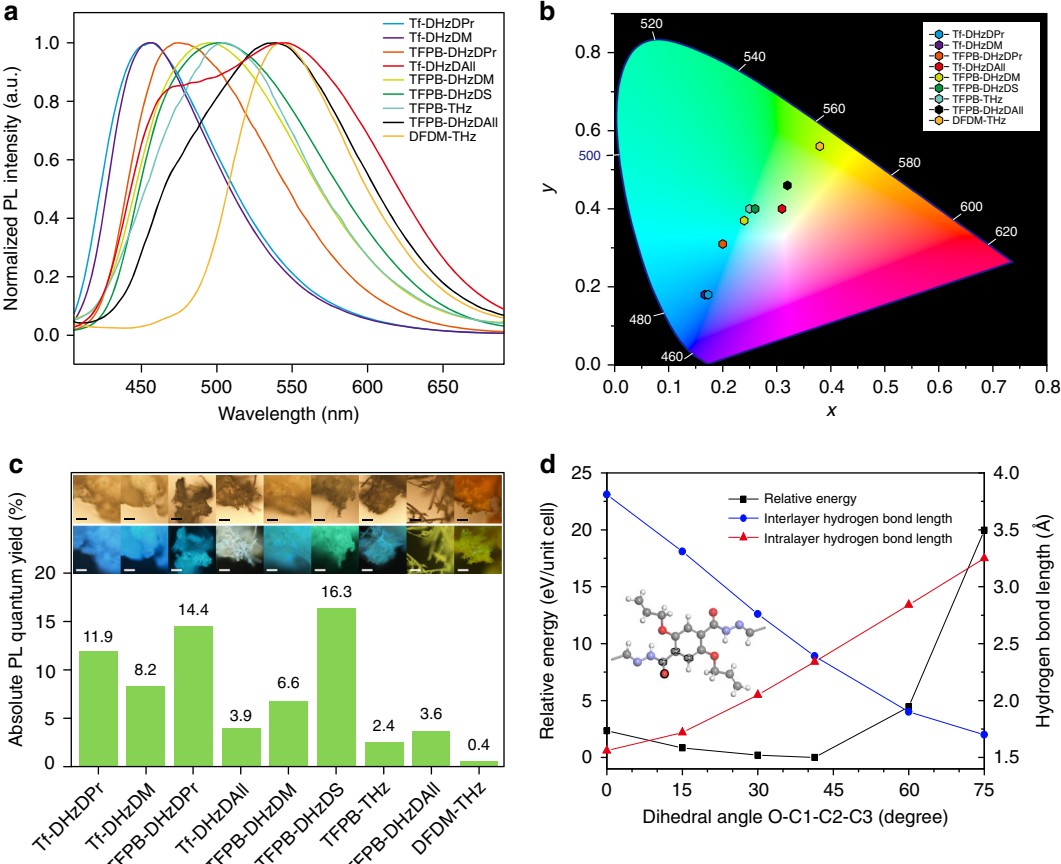

**Fig. 4** Solid-state photoluminescent properties of the tuneable emissive COFs. **a** Solid-state PL spectra. **b** CIE-1931 chromaticity diagram of various COFs. All measurements were performed on solid samples under an excitation wavelength of 365 nm. **c** Absolute PL quantum yields of each COF; inset: fluorescence microscopic images of COFs under visible light (upper) and UV irradiation (down) (scale bar, 50 μm). **d** Torsional potential energy scan of Tf-DHzDAll COF at different dihedral angles; inset: molecular fragment of Tf-DHzDAll COF to show the dihedral angle O-C1-C2-C3 (C, grey; N, violet; O, red; H, white)

432 and 569 m$^2$ g$^{-1}$ for Tf-DHzDPr, TFPB-DHzDPr, Tf-DHzDAll and TFPB-DHzDAll, respectively, suggesting that the four COFs have good to moderate porosity. In addition, the total pore volumes are estimated to be 0.77, 0.63, 0.49 and 0.45 cm$^3$ g$^{-1}$ at $P/P_0 = 0.99$ for Tf-DHzDPr, TFPB-DHzDPr, Tf-DHzDAll and TFPB-DHzDAll, respectively. The pore distributions of the four COFs were calculated by non-local density functional theory (NLDFT), which gave a narrow pore width of 1.69 nm for Tf-DHzDPr and Tf-DHzDAll and a width of 2.77 nm for TFPB-DHzDPr and TFPB-DHzDAll. These values are consistent with the simulated values of 1.74, 1.85, 3.04 and 3.07 nm for Tf-DHzDPr, Tf-DHzDAll, TFPB-DHzDPr and TFPB-DHzDAll, respectively.

The morphologies of each COFs were characterized by Scanning electron microscopy and Transmission electron microscopy (see Supplementary Figs. 17–19).

**Photoluminescent properties of tuneable white-emissive COFs.** As illustrated in Fig. 4c, the nine as-synthesized COF solids display a wide range of PL emission wavelengths from blue, green and white to yellow under UV irradiation. Since 365 nm UV is one of the most commonly available light sources, all the experiments including PL emission and absolute PL quantum yield (PLQY, $\Phi_{PL}$) were conducted at this excitation wavelength unless otherwise specified. To elucidate the relationship between the structures of the COFs and their optical properties, model

compounds were obtained from the condensation of hydrazide and benzaldehyde monomers (Supplementary Fig. 11). The PL emission spectra exhibit maxima at approximately 410–420 nm for the DHzDR models, while a redshift of the maxima to 450 nm was observed for the THz model. Single-crystals of DHzDPr and DHzDAll models were used when measuring their PL spectra and lifetimes. The crystals adopt a herringbone packing arrangement with antiparallel slipped stacking between each molecule (Supplementary Figs. 26–29 and Supplementary Tables 2 and 3). The large distance between each molecule precludes the formation of intermolecular hydrogen bonding, and only intramolecular hydrogen bonding is possible in these crystals. When the hydrazides are bridged with a small spacer such as Tf to form Tf-DHzDR COFs (R = M, Pr, All), the PL spectra show a redshift of approximately 40 nm from their model compounds and form broader emission bands.

Interestingly, the Tf-DHzDAll COF displays a dual emission that is not seen in the corresponding single-crystal model compound and the amorphous Tf-DHzDAll polymer (Fig. 4a and Supplementary Figs. 20 and 21 and Supplementary Table 1). Due to the difficulties in preparing uniform thin film of the materials, we are unable to conduct transient absorption spectroscopy to understand the detailed mechanism of the dual emission. Instead, we utilize DFT simulations to provide insights of mechanism. Dual emission with a large Stokes shift in organic compounds has been attributed to ESIPT[41,42]. According to the ESIPT mechanism, when a molecule is photoexcited, it may

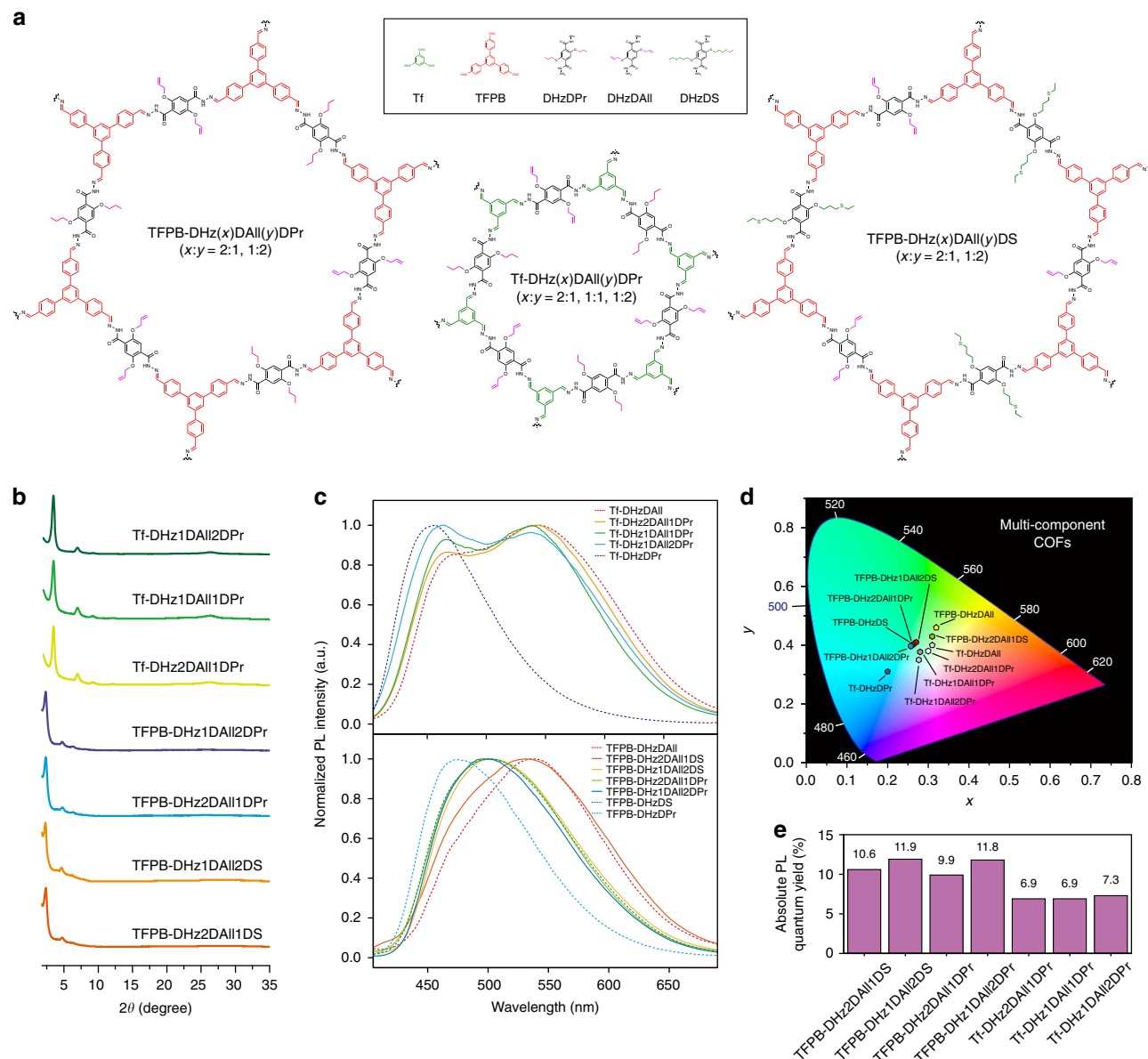

**Fig. 5** Fine-tuning of the PL emission via a multicomponent COF strategy. **a** Synthetic scheme for accessing the multicomponent COFs. **b** PXRD patterns. **c** Solid-state PL spectra. **d** CIE-1991 chromaticity diagram. **e** Absolute PLQYs of various multicomponent COFs

tautomerize (i.e., from the enol to the keto form) through proton transfer, and both tautomers contribute to the dual emission bands. However, in our COF building blocks, the proton acceptor (C–O–C) is not nucleophilic enough to fully capture the hydrogen from the proton donor (N–NH), which disfavours the ESIPT process (refer to DFT simulation of ground and excited states in Supplementary Figs. 34 and 35 and Supplementary Tables 6 and 7). Since neither the single-crystal model compound with herringbone packing (only intramolecular hydrogen bonding) nor the amorphous Tf-DHzDAll polymer (with the building units in a random orientation) show dual emission, we suggest that conformational changes in the excited states due to changes in intra- and interlayer hydrogen bonding may play important roles in the emission process.

To gain insight into how subtle changes in intralayer and interlayer hydrogen bonding affect the fluorescent properties of the COF, we perform first principles DFT calculations using norm-conserving pseudopotentials and a 60 Ry energy cutoff for the plane wave basis set, as implemented in Quantum Espresso

(see Supplementary Methods and Supplementary Note 1)[43]. The lattice parameters were fixed to the parameters mentioned above, while all atoms were relaxed with a force convergence criteria of 0.001 Ry/au. A $1 \times 1 \times 10$ Monkhorst-Pack k-mesh was used. Since the vdW-DF2 exchange-correlation functional has been shown to describe hydrogen bonds well[44–46], we use the vdW-DF2 functional in our calculations. We discover that the intralayer hydrogen bond length is much shorter than the interlayer hydrogen bonding in the single-emissive Tf-DHzDPr COF, whereas their values are close for the dual-emissive Tf-DHzDAll COF (Fig. 1d, Supplementary Figs. 31 and 32 and Supplementary Tables 4 and 5). One reason is that the presence of π-conjugated allyl group enhances the π–π interactions between the sidechains, thus strengthening the interlayer hydrogen bonding. Furthermore, we also study the rotation barrier[47] of the Tf-DHzDAll COF and find that in the ground state, at a relatively small energy cost, the intralayer hydrogen bond length decreases while the interlayer hydrogen bond length increases, by a rotation of the 2,5-substituted phenyl ring (Fig. 4d). Based on

the above facts, we propose a COF-triggered ESIPS process analogous to ESIPT, where the dual emission bands are from two excited-state conformers. Within this proposal, the emission of the blueshifted conformer is primarily influenced by a structure which is similar to what is seen in the ground state, while the emission of the redshift conformer is dominated by a second structure generated by conformational changes via an interlayer proton shift between in-plane and out-of-plane proton acceptors (C–O–C) in the excited state (Supplementary Figs. 30 and 33 and Supplementary Discussion).

When the hydrazides are bridged with a larger spacer such as TFPB, the resulting COF, TFPB-DHzDR, show a redshift of 50–70 nm relative to the emission of the Tf-COFs, which are connected via a smaller spacer. This suggests that extending the conjugation size is effective to redshift PL emission in such COFs. Similarly, the TFPB-DHzDR COFs (R = M, Pr, S) also exhibit broad emission bands in the visible region. The TFPB-DHzDAll COF shows a very broad band with a shoulder at 490 nm instead of a dual-emissive band. This is probably because the first emission is redshifted into the ESIPS emission region due to the extended conjugation via TFPB. To further verify the role of the proton shift in modulating the emission properties, we synthesized two COFs incapable of hydrogen bonding, namely TFPB-THz and DFDM-THz. Although the condensation product formed between trialdehyde and 2,5-unsubstituted terephthalohydrazide (DHz) is more structurally related to DHzDR COFs, we could only obtain amorphous polymer instead of crystalline COF. Thus, we chose TFPB-THz COF and DFDM-THz COF for comparison with hydrogen bonded DHzDR COFs. As expected, they display much narrower single emission bands compared to the DHzDR COFs. Based on these results, the hydrogen bonding between the COF layers is vital to the broadening of the emission band. The interlayer proton shift in the excited state may create numerous energy states, which in turn give rise to the broadened band or even dual emission. The PL lifetime of each emissive COFs was also recorded as shown in Supplementary Figs. 22–24.

A good criterion for judging the fluorescence properties of our material is the absolute PLQY (Fig. 4c and Supplementary Fig. 25). The THz model compound shows negligible emission compared to that of the DHzDR model compounds, which highlights the importance of the RIR mechanism in the latter compounds. By inhibiting bond rotation, the intra- and interlayer hydrogen bonds reduce energy dissipation and lead to stronger emission. Compared to the THz model compound, TFPB-THz COF and DFDM-THz COF exhibit more intense emission with absolute PLQYs of 2.4% and 0.4%, respectively. This enhancement is primarily due to the RIR mechanism in the COF eclipsed stacking structure, which lacks additional contributions from hydrogen bonding interactions. In addition, without hydrogen bonding and crystalline packing structure, amorphous condensate of Tf and DHz displays no PL emission due to the complete absence of RIR. A much higher PLQY (up to 16.3%) is seen in DHzDR COFs where intra- and interlayer hydrogen bonding and eclipse stacking work synergistically to restrict bond rotation. Crystalline Tf-DHzDPr COF and Tf-DHzDAll COF display stronger PL emissions ($\Phi_{PL}$ = 11.9% and 3.9%) than their amorphous counterpart ($\Phi_{PL}$ = 2.1% and 2.1%) (Fig. 4c and Supplementary Fig. 25), which is a further proof that a well-ordered stacking arrangement that allows intra- and interlayer hydrogen bonding is more effective for the RIR mechanism than intramolecular hydrogen bonding alone. According to the formalism of molecular quantum mechanics, changing the frequency of the bonds rotation decides if energy exchange with other energy states is possible. Therefore, further restricting the rotation can slow the rotation rate of the labile bonds and open

channels to other excited energy states that are normally inaccessible.

The CIE-1931 chromaticity diagram reveals the true colour of the emissions of the tuneable emissive COFs (Fig. 4b). By tuning the spacer length and side-chain functionalities, a kaleidoscope of colours can be achieved, such as the blue colour of Tf-DHzDPr and Tf-DHzDM; the cyan colour of TFPB-DHzDPr, the different shades of green of TFPB-DHzDM, TFPB-DHzDS and TFPB-THz; and the yellow colour of DFDM-THz. Tf-DHzDAll and TFPB-DHzDAll, which have the strongest ESIPS emissions, show CIE coordinates of (0.31, 0.4) in the white colour region and (0.32, 0.46) in yellowish green, respectively. These values are near the ideal white point of (0.33, 0.33), which form a basis for us to further fine-tune the emission to reach white colour.

Using these 2D COF platforms, the optical properties can be systematically tuned by modifying the spacer unit and side chains. By extending the conjugation length of the spacer unit used to link the hydrazides species, the emission of the COFs can be tuned from blue to green with a slight increase in the PLQY. Given the same spacer size, the length of the side chain has no effect on the emission colour; however, the PLQY tends to increase when the side chain is lengthened. Most significantly, when the functionality of the side chain is changed from an $sp^3$-hybridized alkyl chain to a π-conjugated allyl group, dual emission can be switched on with a large Stokes shift to give near white emission.

**Multicomponent COF strategy to fine-tune the PL properties**. To achieve broadband PL emission with high PL efficiency, we propose the following strategy for constructing multicomponent COFs (Fig. 5 and Supplementary Figs. 12, 16)[48]. The design and synthesis are focused on two COFs, Tf-DHzDAll and TFPB-DHzDAll, with CIE coordinates of (0.31, 0.4) and (0.32, 0.46) and PLQYs of 3.9% and 3.6%, respectively. For Tf-derived COFs, the spacer is reacted with 2:1, 1:1 and 1:2 mole ratios of DHzDAll and DHzDPr, and the synthesized compounds are referred to as Tf-DHz($x$)DAll($y$)DPr. Similarly, TFPB-COFs are constructed between the spacer TFPB and 2:1 and 1:2 mole ratios of DHzDAll/DHzDPr or DHzDAll/DHzDS. The PXRD spectra confirm the successful synthesis of multicomponent COFs based on the characteristic peaks at 3.46 and 2.39° for the (100) facets of the Tf-COFs and TFPB-COFs, respectively. For Tf-derived multicomponent COFs, the PL emission exhibits a blueshift with an increasing ratio of DHzDPr, giving rise to CIE coordinates of (0.3, 0.38), (0.28, 0.38) and (0.28, 0.35), which are closer to the white point of (0.33, 0.33). Meanwhile, the absolute PLQY improved from 3.9% to 7.3%. For the TFBP-derived multicomponent COFs, instead of giving two emission maxima, the emission from DHzDAll and DHzDS/DHzDPr merged into one peak at approximately 500 nm, which corresponds to an off-white, light-green colour with CIE coordinates of (0.31, 0.43), (0.27, 0.41), (0.26, 0.4) and (0.26, 0.4). Similarly, compared to TFPB-DHzDAll, the PLQY increased from 3.6% to 11.8%.

**Discussion**

Compared to the RIR-emissive crystals of small molecules, a COF has several advantages: 1. Its extended conjugated structure offers COF materials greater colour tuneability than small emissive units. 2. COFs are more resistant to solvents than small-molecule crystals and are thus more stable. 3. Slip-packed crystals of small molecules can have only intramolecular hydrogen bond, while eclipsed-stacked COFs can have both intra- and interlayer hydrogen bonding in close proximity, which forms the basis for ESIPS. Therefore, COFs allow a wide chemical space for the

design of new solid-state emitters for use as photosensitizers, sensors, lighting, displays and supramolecular encryption.

We have demonstrated a strategy to enhance the solid-state PL of COFs by taking advantage of the RIR mechanism. The eclipsed stacking in 2D COFs containing hydrazone units is accompanied by interlayer hydrogen bonding, which provides additional channels for restricting RIR in addition to intralayer hydrogen bonding. Modulating the interlayer and intralayer hydrogen bonding allows access to excited states with different de-excitation pathways. Through this strategy, we observed dual emission in the blue and yellow wavelengths. By varying the spacer size and side-chain functionalities in this COF series, we can tune the intensity and colour of the emission and achieve near white PL emission. Through construction of multi-component COFs, the emission colour can be further fine-tuned. This study provides new insights into how molecular building blocks in a 2D stacked framework, as well as their linkages and functional groups, can be tuned to achieve synergetic interactions for colour-tuneable PL emission, thereby providing the basis for designing new solid-state emitters.

## Methods

**Tf-DHzDAll COF**. A mixture of 2,5-Bis(allyloxy)terephthalohydrazide (DHzDAll) (23 mg, 0.075 mmol) and 1,3,5-triformylbenzene (Tf) (8.1 mg, 0.05 mmol) in mesitylene/1,4-dioxane (v/v = 3:1, 0.8 mL) was sonicated for 10 min in a Schlenk tube (15 mm × 80 mm). Then the mixture was added with acetic acid (50 μL, 6 M), flash frozen at 77 K, and degassed under freeze–pump–thaw for three cycles. The tube was sealed and heated at 120 °C for 3 days. The precipitate was exchanged with anhydrous tetrahydrofuran (THF) (5 mL) for 10 times and dried at 100 °C under vacuum for 8 h to afford a beige colour solid (25.1 mg, 88%).

**Data availability**. All data are available from the authors upon reasonable request.

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

## Acknowledgements

K.P.L. acknowledges NRF-CRP grant "Two Dimensional Covalent Organic Framework: Synthesis and Applications". Grant number NRF-CRP16-2015-02, funded by National Research Foundation, Prime Minister's Office, Singapore. We acknowledge support from the Singapore National Research Foundation, Prime Minister's Office, under its medium-sized centre program. S.Y.Q., J.W. and Y.C. acknowledge support from grant numbers NRF-NRFF2013-07 and MOE2016-T2-2-132.

## Author contributions

X.L. designed and performed most of the experiments including synthesis and characterization of COFs and PL measurements under the supervision of Y.L. and K.P.L. X.L., Q.G., H.-S.X. and G.-H.N. discussed the COF synthesis and characterization. J.W. performed the D.F.T. and T.D.D.F.T. theoretical calculations under the supervision of S.Y.Q. X.L., J.-S.W., K.P.L., S.Y.Q., Y.C. and J.W. discussed the proposed mechanisms for dual emission. Z.-H.C. performed the T.R.E.S. measurement under the supervision of Q.-H.X. W.T. and K.L. collected the T.E.M. image of COFs. X.L. and K.P.L. co-wrote the manuscript with contribution from S.Y.Q. on theoretical discussions.

## Additional information

**Competing interests:** The authors declare no competing interests.

