## [Peer Review File · Nature Communications]

Reviewers' comments:

Reviewer #1 (Remarks to the Author):

The authors prepared a series of solid-state PL of 2D COFs via modulating the interlayer and intralayer hydrogen bonding. Furthermore, by varying the spacer size and side-chain functionalities in this COF series, the PL of COFs can be tuned to white PL emission. The paper describes the design, preparation and mechanism of the solid-state PL 2D COFs. The topic is of great interest while the experiments and results are well presented.

Some minor criticisms

(1) The author said they cannot obtain crystalline COFs constructed with trialdehyde and 2,5-unsubstituted terephthalohydrazide (DHZ). But the TFPB-THz and DFDM-THz are obvious difference with DHzDR COFs. There is a references (Chem.Comm., 2014, 50, 12615—12618.) that prepared COFs with Tp and 2,5-unsubstituted terephthalohydrazide (DHZ). This COF may own more similar properties with DHzDR COFs.

(2) The detail information for the structure of prepared COFs need to provide, including the Space group, unit cell dimensions and fractional main atomic coordinates.

Reviewer #2 (Remarks to the Author):

This manuscript describes a facile strategy to enhance the solid-state photoluminescence (PL) of hydrazone based COFs with tuneable emissive properties from non-emissive building blocks through the restriction of intramolecular bond rotation (RIR) mechanism via intra- and inter-layered molecular hydrogen bonding. The authors have systematically fine-tuned the intensity and color of PL emission, and were able to achieve the nearly white-light emission from their targeted COFs as well as in multi-component COFs by utilizing different size of incorporated building blocks and side-chain functionalities. I found this manuscript interesting as it seems to provide a simple and straightforward approach to the construction of high emissive COFs in solid-state and their conclusions might find a new generation of highly luminescent COF materials that retain high quantum yields in both solid and solutions and function as highly sensitive sensor to detect specific chemicals.

There is a major concern on the white light emission concept, if the material supposed to be white-light emission it should be covered their emission spectrum in the entire visible range such as blue, green and red regions more or less in equal proportions (0.33, 0.33, 0.33 in the CIE-1931 chromaticity diagram). However, the obtained CIE coordinates values of reported two best emissive COFs Tf-DHzDAll (0.31, 0.40, 0.39), TFPB-DHzDAll (0.32, 0.46, 0.22) are not that close to the ideal white-light emissive materials. In that case, it is not appropriate to give the statement as "These are very close to the white point of (0.33, 0.33) and can be considered white light emission". Additionally, the visible images of above mentioned COF (TFPB-DHzDAll) display yellow color domination in their PL emission instead of white (in Figure 4a). Further, there is a discrepancy between solid-state PL spectra (or CIE-1931 diagram values, Figure 4b,c) of various COFs and observed PL emission shown in images (irradiated under visible light and 365 nm UV irradiation, Figure 4a), not much distinguish in the emission observed although CIE-1931 and PL spectral displayed clear difference. Therefore, I would recommend the authors to use the laser confocal fluorescence microscope for displaying the true emission from all samples under same excitation.

Few additional comments:

- (1) FT-IR analysis in the supplementary information is too confusing. It either splits in to 3-4 figures or gives the corresponding chemical structure besides for better understanding for the readers.
- (2) Authors should provide more details on the sample preparations, loaded amount and internal standards (if there is any), etc. for all analytics for example, photoluminescence spectra, lifetime and absolute quantum yield calculations, SEM, TEM, and so on.
- (3) Fitting/simulated BET surface area plot of for all the COFs should be given and if possible also mention the value of coefficient of determination.

Reviewer #3 (Remarks to the Author):

The MS reports a novel strategy to construct covalent organic frameworks (COFs) that enable photoluminescence (PL) in the solid state. By capitalizing on the flexibility of the linker chromospheres and hydrogen-bond-mediated "freezing" of rotational degrees of freedom, the authors succeeded to color-tune their COFs and even generate white emission from a mixture of appropriately chosen COFs (see Fig. 2). High PL from COFs had been reported previously, as for instance mentioned in refs. 19, 36-38 of the MS, but quenching in COFs containing often-used boronate esters or imine groups in their linkers occurs sometimes due to internal rotational degrees of freedom. Note that ref. 19 solved this problem by appropriate dense design of pi-stacking, while

the present MS reports "freezing" these modes that cause internal conversion and non-radiative decay by introducing hydrogen bonds to maintain the planarity of the singly-bonded linkers. In this sense, the work reports a new design strategy, although the individual strategies such as hydrogen-bonded freezing has also already reported before (see for instance X. Chen et al, "Locking Covalent Organic Frameworks with Hydrogen Bonds: General and Remarkable Effects on Crystalline Structure, Physical Properties, and Photochemical Activity", J. Am. Chem. Soc. 137, 9, 3241-3247, please insert appropriate citations), and the PL from COFs was also reported before. The versatility of the available linker groups allows the wide range for color tuning, in the end achieving the "holy grail" of white light PL from a mixture of COFs. Therefore, this contribution deserves broad attention in the field, and publication as an article in Nature Communications is recommended after the following minor issues are appropriately addressed.

Being a theoretician, I will restrict my comments to the theory parts only.

1) In the supporting information, when the crystal structure geometry optimization using DFT is reported, it is essential to add the details. What code was used, which functional, what plane wave basis set/pseudopotentials, what was the energy cutoff, what were the convergence thresholds, etc.

2) It would be great if the authors could provide torsional potential energy scans at least in the electronic ground state, such as for instance shown in Chemical Physics Letters 664 (2016) 101–107. Incidentally, this reference should be cited in the paper as it is highly relevant to the presented problem of locking the COF planar structure.

Point by point response to reviewers' comments for NCOMMS-18-06524, "Tuneable, White-emissive Two-Dimensional Covalent Organic Frameworks" by Xing Li et. al.

Reviewer 1

We like to thank the referee for the positive endorsement of our work and our responses are included below.

(1) The author said they cannot obtain crystalline COFs constructed with trialdehyde and 2,5-unsubstituted terephthalohydrazide (DHZ). But the TFPB-THz and DFDM-THz are obvious difference with DHzDR COFs. There is a references (Chem.Commun., 2014, 50, 12615—12618.) that prepared COFs with Tp and 2,5-unsubstituted terephthalohydrazide (DHZ). This COF may own more similar properties with DHzDR COFs.

ANS: We have synthesized the TpTh COF using solvothermal method according to Banerjee's work (Response Fig. 1).¹ The TpTh COF is weakly emissive in orange colour with absolute quantum yield of 0.8%. The weak PL is caused by the restriction of intramolecular bond rotation (RIR) via eclipse stacking of the COF. However, due to the absence of 2,5-dialkyloxy group at the hydrazide units, the TpTh COF is incapable of forming intralayer and interlayer hydrogen bonding for RIR, thus it shows weak PL. This example is similar to TFPB-THz and DFDM-THz COFs prepared in our work, which also lack intra- and interlayer hydrogen bonding and display relatively weak PL with absolute quantum yield of 2.4 % and 0.4%, respectively. These results suggest that the intra- and interlayer hydrogen bonding realised by the COF's eclipse stacking is vital to achieve strong PL emission in such materials.

Response Figure 1 | Characterization and PL properties of TpTh COF. **a**, Powder XRD pattern of TpTh COF. **b**, chemical structure of TpTh COF. **c**, images of TFPB-DHzDS COF and TpTh COF under visible light and UV irradiation at 365 nm with absolute PL quantum yields. **d**, Absorption (blue) and emission (red) spectra of TpTh COF.

(2) The detail information for the structure of prepared COFs need to provide, including the Space group, unit cell dimensions and fractional main atomic coordinates.

ANS: The required data is added in Section Q in supplementary information. Note that for structure determination, the crystal structure was optimized using the Materials Studio Forcite molecular dynamics module with ultra-fine, Universal force fields, Ewald summations condition, and then Pawley refinement was performed using the pseudo-Voigt profile function and Berrar–Baldinozzi function for whole profile fitting and asymmetry correction

respectively. To further gain insight of the hydrogen bonding environment in the COFs, more accurate models were relaxed and optimized from the previously refined structures using Density Functional Theory (DFT) with fixed cell parameters but a lower space group (*P1*). (See detail in structural modelling method in section A and DFT optimized crystal structure in section N in the supplementary information.)

Reviewer 2

There is a major concern on the white light emission concept, if the material supposed to be white-light emission it should be covered their emission spectrum in the entire visible range such as blue, green and red regions more or less in equal proportions (0.33, 0.33, 0.33 in the CIE-1931 chromaticity diagram). However, the obtained CIE coordinates values of reported two best emissive COFs Tf-DHzDAII (0.31, 0.40, 0.39), TFPB-DHzDAII (0.32, 0.46, 0.22) are not that close to the ideal white-light emissive materials. In that case, it is not appropriate to give the statement as “These are very close to the white point of (0.33, 0.33) and can be considered white light emission”. Additionally, the visible images of above mentioned COF (TFPB-DHzDAII) display yellow color domination in their PL emission instead of white (in Figure 4a). Further, there is a discrepancy between solid-state PL spectra (or CIE-1931 diagram values, Figure 4b,c) of various COFs and observed PL emission shown in images (irradiated under visible light and 365 nm UV irradiation, Figure 4a), not much distinguish in the emission observed although CIE-1931 and PL spectral displayed clear difference. Therefore, I would recommend the authors to use the laser confocal fluorescence microscope for displaying the true emission from all samples under same excitation.

ANS: Thank you very much for the concerns and suggestions. In line with the suggestion of the referee, we have decided to change the description to “near white” to differentiate from “white”.

Ideally, pure white light should be at (0.33, 0.33, 0.33) CIE coordinates. However, most white-emissive materials deviate from this ideal value. For example, a white-emissive perovskite was reported with CIE coordinates of (0.28, 0.36).² A review on white-light-emitting perovskites includes various examples with CIE coordinates even such as (0.39, 0.42), (0.23, 0.29) and etc.³ Another report showed white light emission of a hydrocarbon

nanoring-iodine assembly with CIE coordinate of (0.26, 0.38).⁴ To the best of our knowledge, currently there is no strict definition of white light emission in colour science. As shown in Response Fig. 2, the CIE diagram (<http://hyperphysics.phy-astr.gsu.edu/hbase/vision/cie.html>) reveals different colour regions, which white colour is in the middle cycle. As long as the CIE coordinates fall into this region, we consider it as white emission. We have summarized all the CIE coordinates of COFs into Response Table 1. COFs of Tf-DHzDAll, Tf-DHz2DAll1DPr, Tf-DHz1DAll1DPr and Tf-DHz1DAll2DPr are all in the white region in Response Fig. 2. Although they are not ideally white, we can still consider them as white-emissive. TFPB-DHzDAll COF is out of the white region and near to the cross section between white and yellow green, so it should indeed be considered as yellowish colour. Therefore, we change the statement to “These values are near the ideal white point of (0.33, 0.33), which form a basis for us to further fine-tune the emission to reach white colour.”

Response Figure 2 | A CIE diagram which defines the colour regions.

Response Table 1 | Summary of CIE Coordinates of COFs.

COF	CIE coordinates
Tf-DHzDPr	(0.17, 0.18)
Tf-DHzDM	(0.17, 0.18)
TFPB-DHzDPr	(0.2, 0.31)
TFPB-DHzDM	(0.24, 0.37)
TFPB-DHzDS	(0.26, 0.4)
TFPB-THz	(0.25, 0.4)
DFDM-THz	(0.38, 0.56)
Tf-DHzDAII	(0.31, 0.4)
TFPB-DHzDAII	(0.32, 0.46)
Tf-DHz2DAII1DPr	(0.3, 0.38)
Tf-DHz1DAII1DPr	(0.28, 0.38)
Tf-DHz1DAII2DPr	(0.28, 0.35)
TFPB-DHz2DAII1DS	(0.31, 0.43)
TFPB-DHz1DAII2DS	(0.27, 0.41)
TFPB-DHz2DAII1DPr	(0.26, 0.4)
TFPB-DHz1DAII2DPr	(0.26, 0.4)

ANS: We do not possess a laser confocal fluorescence microscope. Instead, we used a fluorescence microscope with a mercury lamp (UV region) as the excitation source. As shown in revised Fig. 4 in the main text and response Fig. 3, the fluorescence colour of each COF is closer to the PL spectra and chromaticity diagram under the fluorescence microscope.

Response Figure 3 | Fluorescence microscopic images of COFs.

(1) FT-IR analysis in the supplementary information is too confusing. It either splits in to 3-4 figures or gives the corresponding chemical structure besides for better understanding for the readers.

ANS: FT-IR spectra figures are redrawn as supplementary Fig. 10-13 with corresponding chemical structures beside.

(2) Authors should provide more details on the sample preparations, loaded amount and internal standards (if there is any), etc. for all analytics for example, photoluminescence spectra, lifetime and absolute quantum yield calculations, SEM, TEM, and so on.

ANS: The samples were stored in dry desiccator unless specified and used as-synthesized without further treatment for each analytic measurements. No internal standards were used for each measurements.

For PL spectra and lifetime measurement, the data were recorded on a Horiba Fluorolog-3 spectrofluorometer equipped with a FluoroHub R-928 detector. The PL was measured using excitation wavelength of 365 nm from a Tungsten lamp. Lifetimes were recorded using a 374 nm nanoLED as excitation source. Diluted LUDOX[®] HS-40 colloidal silica was used for lifetime prompt measurement. The loaded amount of COF was around 5 mg to fill the sample holder as shown in Response Fig. 4a.

The absolute PL quantum yield determination were recorded on the HORIBA Fluorolog-3 Photon Counting Spectrofluorometer System with Quanta- ϕ 6-inch integrating sphere. 15 mg of COF samples or model compounds was loaded onto the holder (Response Fig. 4b) to fully cover the bottom.

For SEM, the COFs were dispersed in ethanol and drop-casted onto silica substrates. After drying, the substrates were sputtered with ~9 nm platinum for measurement. Similarly, the COF dispersions in ethanol were drop-casted to SPI copper grid (200-mesh, holey carbon) and dried for TEM measurement.

ACTION: all these descriptions have been added in SI, under the general information section A.

Response Figure 4 | Holders for measurements. a, PL and lifetimes. **b,** Absolute PL quantum yield.

(3) Fitting/simulated BET surface area plot of for all the COFs should be given and if possible also mention the value of coefficient of determination.

ANS: The fittings of multipoint BET surface area plots with correlation coefficient (r) have been added into supplementary Fig. 1-9.

Reviewer 3

Thank you very much for the comments and suggestions.

ANS: We have acknowledged previous contribution where intralayer hydrogen bonding was involved in synthesizing highly crystalline COFs for photocatalysis application and cited it in reference 25.

1) In the supporting information, when the crystal structure geometry optimization using DFT is reported, it is essential to add the details. What code was used, which functional, what plane wave basis set/pseudopotentials, what was the energy cutoff, what where the convergence thresholds, etc.

Details provided as follow:

ANS: To further understand the hydrogen bonding environment in the COFs, more accurate models were relaxed and optimized from the previously refined structures using the plane wave Density Functional Theory (DFT) code, Quantum Espresso.⁵ During relaxation, we fixed the unit cell shape and size with lattice constants, $a = b = 29.37 \text{ \AA}$, $c = 3.44 \text{ \AA}$ for Tf-DHzDPr, and $a = b = 29.35 \text{ \AA}$, $c = 3.43 \text{ \AA}$ for Tf-DHzDAII. Optimizing these lattice constants did not change the results significantly. We used a norm-conserving PBE pseudopotential, with a 60 Ry kinetic energy cutoff, a $1 \times 1 \times 10$ Monkhost-Pack k-mesh, and a 10^{-6} Ry convergence threshold for self-consistency. For geometry optimization, we used a 10^{-4} Ry convergence threshold on total energy and a 10^{-3} Ry/au convergence threshold on forces. The vdW-DF2 method was utilized to take the long-range van der Waals interactions into account.

2) It would be great if the authors could provide torsional potential energy scans at least in the electronic ground state, such as for instance shown in Chemical Physics Letters 664 (2016) 101–107. Incidentally, this reference should be cited in the paper as it is highly relevant to the presented problem of locking the COF planar structure.

ANS: We have included the torsional potential energy scan in the electronic ground state in Fig. 4d and Response Fig. 5 using the dual-emissive Tf-DHzDAII COF as an example. It suggests that the hydrogen bonding indeed restricts the rotation of intramolecular bonds and keeps the COF planar due to the large energy barrier when the dihedral angle O-C1-C2-C3 is bigger than 50° . Meanwhile, at relatively small energy cost, the intralayer hydrogen bond length can decrease while the interlayer hydrogen bond length can increase. This suggests

that the interplay between the intra- and interlayer hydrogen bonding can provide a channel for proton shift *via* partial bond rotation to access different emissive energy state, leading to dual emission in Tf-DHzDAII COF.

ANS: We have added the literature as reference 47 in the main text.

Response Figure 5 | Rotation barrier study of Tf-DHzDAII COF. a, Torsional potential energy scan of Tf-DHzDAII COF at various dihedral angles. **b,** A unit cell of Tf-DHzDAII COF with labelled dihedral angle O-C1-C2-C3 (C, grey; N, violet; O, red; H, white).

Reference

1. Das, G., Balaji Shinde, D., Kandambeth, S., Biswal, B. P. & Banerjee, R. Mechanosynthesis of imine, β -ketoenamine, and hydrogen-bonded imine-linked covalent organic frameworks using liquid-assisted grinding. *Chem. Commun.* **50**, 12615-12618 (2014).
2. Mao, L., Wu, Y., Stoumpos, C. C., Wasielewski, M. R. & Kanatzidis, M. G. White-Light Emission and Structural Distortion in New Corrugated Two-Dimensional Lead Bromide Perovskites. *J. Am. Chem. Soc.* **139**, 5210-5215 (2017).
3. Smith, M. D. & Karunadasa, H. I. White-Light Emission from Layered Halide Perovskites. *Acc Chem Res.* **51**, 619-627 (2018).
4. Ozaki, N. *et al.* Electrically Activated Conductivity and White Light Emission of a Hydrocarbon Nanoring–Iodine Assembly. *Angew. Chem., Int. Ed.* **56**, 11196-11202 (2018).
5. Paolo, G. *et al.* QUANTUM ESPRESSO: a modular and open-source software project for quantum simulations of materials. *J. Phys. Condens. Matter* **21**, 395502 (2009).

REVIEWERS' COMMENTS:

Reviewer #1 (Remarks to the Author):

The revised manuscript has addressed my previous comments. I have no further comments.

Reviewer #2 (Remarks to the Author):

I am glad to see the authors' response and efforts on proper addressing to my review comments/suggestions. I am reasonably convinced with their valid justifications and changes that are made in the draft as well as in the ESI, such as incorporation of updated fluorescence microscope images in Figure 4, FT-IR, BET and analytical details on the sample preparations and measurements (in ESI), and the title "Near white-emissive" (instead of white light), etc. Therefore, I would recommend the publication of this manuscript in the 'Nature Communications' as it is.

Reviewer #3 (Remarks to the Author):

The revision produced by the authors is almost satisfactory. In particular appreciated are the author's additional efforts on the calculation of the torsional potentials, illustrating the importance of inter- as well as intra-layer hydrogen bonding for the torsional dynamics, now shown in revised Figure 4.

One detail remains to be addressed, and that is the name of the exchange functional. DFT contains two functionals, not one, an exchange functional (for instance revised PBE - sometimes denoted revPBE or RPB), and a correlation functional (the authors mention they used the vdW-DF2 correlation functional implemented in Quantum Espresso). The authors need to specify which exchange functional was used in conjunction with vdW-DF2. If a default choice was accepted, the authors need to figure out what is the default in their particular quantum chemistry code of choice. Presumably it is revPBE but this needs to be verified.

We like to thank the referees for the positive endorsement of our work.

Reviewer 3

One detail remains to be addressed, and that is the name of the exchange functional. DFT contains two functionals, not one, an exchange functional (for instance revised PBE - sometimes denoted revPBE or RPB), and a correlation functional (the authors mention they used the vdW-DF2 correlation functional implemented in Quantum Espresso). The authors need to specify which exchange functional was used in conjunction with vdW-DF2. If a default choice was accepted, the authors need to figure out what is the default in their particular quantum chemistry code of choice. Presumably it is revPBE but this needs to be verified.

ANS: The exchange functional we used was PBE according to Supplementary Reference 11. [Perdew, J.P., Burke, K. & Ernzerhof, M. Generalized Gradient Approximation Made Simple. *Phys. Rev. Lett.* **77**, 3865-3868 (1996)]

We have added the detail of vdW-DF2 into Supplementary Note 1. More details of vdW-DF2 are given in Supplementary Reference 12. [Lee, K., Murray, E. D., Kong, L., Ludqvist, B. I. & Langreth D. C. Higher-accuracy van der Waals density functional. *Phys. Rev. B* **82**, 081101 (2010)]